# Associations between Anxiety, Depression, Chronic Pain and Oral Health-Related Quality of Life, Happiness, and Polymorphisms in Adolescents’ Genes

**DOI:** 10.3390/ijerph20043321

**Published:** 2023-02-14

**Authors:** Ana Luiza Peres Baldiotti, Gabrielle Amaral-Freitas, Mariane Carolina Faria Barbosa, Paula Rocha Moreira, Renato Assis Machado, Ricardo Della Coletta, Michelle Nascimento Meger, Saul Martins Paiva, Rafaela Scariot, Fernanda de Morais Ferreira

**Affiliations:** 1Department of Pediatric Dentistry, Federal University of Minas Gerais, Belo Horizonte 31270-901, MG, Brazil; 2Department of Morphology, Institute of Biological Sciences (ICB), Federal University of Minas Gerais, Belo Horizonte 31270-901, MG, Brazil; 3Department of Oral Diagnosis and Graduate Program in Oral Biology, School of Dentistry, University of Campinas, Piracicaba 13414-903, SP, Brazil; 4School of Health and Biological Sciences, Universidade Positivo, Curitiba 81290-000, PR, Brazil; 5Departament of Oral Surgery and Maxilofacial, Federal University of Paraná, Curitiba 81530-000, PR, Brazil

**Keywords:** adolescent, happiness, anxiety, depression, chronic pain, quality of life, genetic polymorphisms

## Abstract

Adolescence is marked by changes and vulnerability to the emergence of psychological problems. This study aimed to investigate associations between anxiety/depression/chronic pain and oral health-related quality of life (OHRQoL)/happiness/polymorphisms in the *COMT*, *HTR2A* and *FKBP5* genes in Brazilian adolescents. A cross-sectional study was conducted with ninety adolescents 13 to 18 years. Anxiety, depression and chronic pain were evaluated using the RDC/TMD. The Oral Health Impact Profile was used to assess oral OHRQoL. The Subjective Happiness Scale was used to assess happiness. Single-nucleotide polymorphisms in *COMT* (rs165656, rs174675), *HTR2A* (rs6313, rs4941573) and *FKBP5* (rs1360780, rs3800373) were genotyped using the Taqman^®^ method. Bivariate and multivariate logistic regression analyses were performed (*p* < 0.05). Chronic pain and depression were associated with feelings of happiness (*p* < 0.05). A significant inverse association was found between anxiety and OHRQoL (*p* = 0.004). The presence of minor allele C of *COMT* rs174675 was significantly associated with depression (*p* = 0.040). Brazilian adolescents with depression and chronic pain considers themselves to be less happy than others and those with anxiety are more likely to have a negative impact on OHRQoL. Moreover, the rs174675 variant allele in the *COMT* gene was associated with depressive symptoms in Brazilian adolescents.

## 1. Introduction

Adolescence is a transitional period of human development between childhood and adulthood marked by physical, pubertal and emotional interactions that may predispose individuals to the onset of internalizing disorders, such as anxiety and depression [1,2]. These internalizing disorders are the most frequent mental health problems found in children and adolescents [3]. The pathophysiology of these conditions is not yet fully understood and treatment results are generally unsatisfactory [4].

For many years, studies were focused on the conditions’ physiopathology, which is extremely important to understanding and treating disease. However, more recently, it was also acknowledged the importance of the Patient-Related Outcome Measure (PROMs), which collects with questionnaires health outcomes directly from people who experience them. This is imperative to recognize how the disease has affected people’s lives, and that could be a key to successful interventions and treatments [5,6].

The oral health-related quality of life (OHRQoL) is a patient-centered subjective measure, which represents the patient’s oral health status [7]. Currently, studies have shown associations between the impact of OHRQoL and probable depression [8,9] and anxiety [8,10], and this is a bilateral association. Therefore, we hypothesize that the negative impact on the OHRQoL occurs because of a worse oral health condition, and that this could also have an impact on mental condition.

Although there are many definitions, happiness is a subjective assessment of well-being, how people face their life, experiences, and the evolution of life as a whole [11,12]. There are many variables involved in this construct [13,14] and it has been described that depressed individuals avoid experiencing their happiness for fear of resulting in negative emotions [11]. Therefore, aspects related to happiness may influence mental health.

Studies on the etiology of emotional disorders have demonstrated the role of genetic variants, which are likely due to a combination of environmental factors and multiple genetic polymorphisms [15,16]. Moreover, substantial overlap is believed to occur between genes that affect psychological conditions [17,18]. Studies have investigated the association between anxiety and depression with polymorphisms in candidate genes involved in the neurotransmitter system and the stress response [19], such as Catechol-O-methyltransferase (*COMT)* [20,21,22], Serotonin receptor 2A (*HTR2A)* [23] and FK506-binding protein 51 (*FKBP5)* [24,25].

*COMT* metabolizes catechol neurotransmitters such as dopamine, noradrenaline, and adrenaline. Those are involved in pivotal brain and physiological functions including mood, cognition, pain, and stress response [20,21]. Moreover, the *HTR2A* has a key role in the serotonin pathway regulation, which forms a part of cation channels. This gene has a complex role that mediates an excitatory effect. Therefore, can play a role in the pathophysiology, and time course of treatment response of psychiatric disorders [23]. Finally, *FKBP5* has an inhibitory role in the modulation of glucocorticoid receptor signaling with intracellular as well as systemic effects [24,25]. Thus, these genes may be associated with emotional disorders and psychopathologies.

However, few studies have evaluated these variables concurrently in adolescents [26]. Moreover, evidence suggests that chronic pain may be related to anxiety and depression [27,28,29], as a high frequency of psychological problems is found in individuals with chronic pain [30,31]. Although, we did not find studies that evaluated these three outcomes together. Therefore, in this study we investigated the association between anxiety/depression/chronic pain and OHRQoL/feelings of happiness/polymorphisms in the COMT, HTR2A, and FKBP5 genes in Brazilian adolescents.

## 2. Materials and Methods

This cross-sectional study was conducted in accordance with *STrengthening the REporting of Genetic Association studies* (STREGA statement) (https://www.equator-network.org/reporting-guidelines/strobe-strega/, accessed on 2 February 2023) and received approval from the Human Research Ethics Committee of the Federal University of Minas Gerais (Protocol #01936918.8.0000.5149). Parents/guardians and adolescents received written clarifications regarding the study and signed statements of informed consent.

The sample was composed of all adolescents who attended the Federal University of Minas Gerais Dental Clinic between May and December 2019 and who met the eligibility criteria. It was included in this study adolescents of both sexes, biologically unrelated, 13 to 18 years old. Excluded were those adolescents currently undergoing orthodontic treatment, using dental prostheses, experiencing odontogenic pain, with severe facial or dental anomalies, with parental reports of cognitive or behavioural problems, or with systemic disorders.

The data collection instruments were administered by two examiners who had undergone training and calibration exercises supervised by professionals experienced in the use of these instruments. The examiners received theoretical and practical training involving interviews with 28 patients who were not included in the main study. A pilot study was conducted with 10 adolescents to test the methods. There was no need to change the proposed methods and these individuals were included in the main study.

The clinical exam was conducted in the dental chair, under artificial light. Personal protective equipment and a sterile clinical kit consisting of a mouth mirror and WHO dental probe was used during the examination of all patients. The self-administered questionnaires were applied to the adolescents in a private space with a desk, chair, pen, and artificial light.

Chronic pain was diagnosed using the validated Brazilian version of the Research Diagnostic Criteria for Temporomandibular Disorders (RDC/TMD)—Axes I [32] and II [33]. The diagnosis of anxiety and depression was performed using Axis II of the RDC/TMD. 

Axis I of the RDC/TMD is an objective clinical assessment for the diagnosis of muscle disorders, disc displacement and joint disorders related to temporomandibular disorder. Axis II of the instrument constitutes a self-report questionnaire with 31 questions. Question 20 of this axis is a psychometric scale consisting of 32 items for the assessment of psychosocial functioning and pain-related disability. The score for each item ranges from 0 to 4 points. The mean is calculated. Depression and nonspecific physical symptoms, including generalized anxiety disorder and pain, are classified as normal, moderate or severe [34,35].

Although the RDC/TMD provides several diagnoses, only diagnoses of chronic pain (obtained through the combination of Axis I and II), depression and anxiety (measured using Axis II) were considered in the present investigation. These variables were dichotomized as present or absent. The Diagnostic Criteria for Temporomandibular Disorders (DC/TMD) was not used because it had not been validated in Brazilian Portuguese at the time of data collection.

OHRQL was evaluated using the validated Brazilian version of the Oral Health Impact Profile (OHIP-14) [36], which has 14 questions distributed among 7 domains. Questions have 5 answer options ranging from 0 (never) at 4 (often). The total ranges from 0 to 56 points, with higher scores denoting greater impact [37,38]. The results were dichotomized into “absence of negative impact” (0 = never and 1 = hardly ever) or “presence of negative impact” on OHRQoL (2 = occasionally, 3 = fairly often and 4 = often) [39,40]. The same reasoning was employed for the interpretation of the separate domains.

The participants answered the validated Brazilian version of the Subjective Happiness Scale (SHS) [40,41], which measures subjective global happiness through self-reporting. The instrument address whether the respondent considers herself or himself to be a happy or unhappy person based on ratings attributed to four affirmative items where the respondent indicates their happiness through a visual analog scale with seven points. Each item has a scale in ascending order of happiness from one to seven, of which the lower scores denote a lower degree of happiness and higher scores a higher degree of happiness [40,41]. In the present study, the total score (ranging from 0 to 28) was used to analyze the data. 

All instruments used in this research were self-administered and followed their original usage methods.

In addition, the adolescents were asked about their age, sex, date of birth, and the presence of headaches. The reason for collecting these variables was because the literature indicates that they could influence the study outcomes. 

DNA was collected for genotyping following the protocol established by Kuchler et al. (2012) [42]. The selection of polymorphisms was based on previous candidate-gene identification studies [9,12,13,14,15,16,23]. Table 1 describes gene characterization, polymorphisms, position, minor allele frequency (MAF) and altered base in general population. Polymorphisms in the *COMT* (rs165656, rs174675), *HTR2A* (rs6313, rs4941573) and *FKBP5* (rs1360780, rs3800373) genes were investigated using polymerase chain reaction (PCR) with the TaqMan^®^ method in a real-time PCR system (Applied Byosistems^®^, 7500 Real-Time PCR System, Thermo Fisher Scientific, Foster City, CA, USA).

Microsoft Excel and the Statistical Package for the Social Sciences (SPSS, version 22.0, IBM Corp., Armonk, NY, USA) were used. Descriptive analysis was performed first. Bivariate analyses were then performed between the outcome and variables of interest considering an underlying theoretical framework. Unadjusted and adjusted binary logistic regression models were run to assess associations between chronic pain/anxiety/depression and OHRQoL/feelings of happiness/genetic polymorphisms. Enter method was used to create the final model, generating adjusted odds ratios (OR) and respective 95% confidence intervals (CI) for anxiety, depression, chronic pain and the independent variables. The alpha was 5%. 

This study was approved by the Human Ethics Committee of Federal University of Minas Gerais (Protocol #01936918.8.0000.5149). Informed written consent was obtained from the Parents/caregivers and those adolescents 18 years old. Assent document was used for all adolescents under 18 years.

## 3. Results

Ninety adolescents participated in the study; however, 105 adolescents were invited to participate in this study (response rate: 84.8%) as seen in Figure 1. Forty-six participants were girls (51.7%) and forty-four were boys (48.3%). The mean age was 15.9 years (SD = 1.66).

Table 2 displays the frequencies of each outcome. The most frequent condition was depression (48.3%), followed by anxiety (42.7%) and chronic pain (28.9%). 

In the univariate analysis (Table 3), the SHS score was associated with all three outcomes: chronic pain (*p* = 0.016), depression (*p* = 0.002) and anxiety (*p* = 0.040). All three main variables were also associated with negative impact on OHRQoL: chronic pain (*p* ≤ 0.001), depression (*p* = 0.008) and anxiety (*p* = 0.001). Depression was significantly associated with sex (*p* = 0.001), with women having more depression than men. No associations were found between the outcomes and genetic polymorphisms in the additive model.

The adjusted multivariate regression model for anxiety (Table 4) revealed an association with negative impact on OHRQoL (*p* = 0.004).

The adjusted multivariate regression model for depression (Table 5) revealed associations with the SHS score (*p* = 0.032) and the genotype CC of rs174675 (*p* = 0.040).

The adjusted multivariate regression model for chronic pain (Table 6) revealed an association with the SHS score (*p* = 0.042).

## 4. Discussion

This study evaluated anxiety, depression and chronic pain in adolescents as well as associations with oral health-related quality of life, feelings of happiness and polymorphisms in the *COMT*, *HTR2A*, and *FKBP5* genes. Associations between anxiety, depression and chronic pain are well described in the literature [8,43,44]. However, to the best of our knowledge, this is the first study that simultaneously evaluated the association of all these aspects in adolescents. Our hypothesis is that OHRQoL, feelings of happiness, and polymorphisms in candidate genes, such as *COMT*, *HTR2A* and *FKBP5*, could be associated with anxiety, depression and chronic pain.

Adolescence is a period of human development involving physiological changes and maturational advances in cognitive, social, and affective capacities [1]. This period is characterized by sensation-seeking, increased risk-taking, and sensitivity to social judgments [1,2]. The beginning of adolescence, which is the transitional period from childhood, constitutes a key developmental window for understanding the risks of mental disorders in youths and, therefore, has enormous implications for public health [1]. A recent cohort study in Denmark found a 15.01% risk of mental disorders before the age of 18 years [45] and a systematic review found that the overall prevalence of mental disorders in young people was 13.4%, with an emphasis on anxiety and depression disorders [3]. 

Mental disorders can exert an impact on general quality of life [46] as well as OHRQoL [8,9,10]. OHRQoL is a multidimensional concept involving the subjective assessment of physical, psychological and social aspects related to oral health. In the present study, anxiety, depression and chronic pain were associated with a negative impact on OHRQoL in the bivariate analyses. However, when evaluated together with other factors in the multivariate analysis, only the association with anxiety was sustained. The impacts of anxiety [8,47], depression [8,44,47] and chronic pain [43,44] on OHRQoL are well described in the literature. However, the studies cited were with adults, children and pre-adolescents rather than adolescents. To the best of our knowledge, this is the first study to perform such an analysis on the adolescent population. 

Subjective measures have been increasingly employed to gain a better understanding of the development, prognosis, and treatment of adverse health conditions as well as the meaning individuals attribute to these conditions. Such measures address psychological, social, emotional and functional domains [48]. 

Happiness is a subjective measure and its concept varies greatly between cultures and age groups. In general, it is how individuals judge their life, which can be the result of their choices, opportunities and experiences, including emotional and cognitive domains [13,14]. Numerous factors are linked to happiness, including mental and physical health. Therefore, we decided to evaluate possible associations between happiness and chronic pain, depression and anxiety. Associations were found in the bivariate analyses. To the best of our knowledge, this is the first study to test such associations. When adjusted by other variables, the Subjective Happiness Scale score remained associated with chronic pain and depression, as adolescents with these conditions (concomitantly or separately) reported being less happy. These findings are consistent with the results of an important study involving 24,118 adults, which concluded that individuals who were less healthy considered themselves to be less happy and vice-versa [49]. Therefore, the findings lend strength to our hypothesis that the subjective measure of happiness is directly related to aspects of physical and mental health in Brazilian adolescents. However, studies with larger samples of adolescents are needed to confirm this association.

In recent years, the investigation of the role of genetic factors related to anxiety, depression and pain has focused on genes of the serotonin pathways [15,21,23] and pro-inflammatory cytokines [19,50]. The selection of genetic polymorphisms in the present study was based on characteristics reported in the literature regarding the role of genes [51]. We found an association between the rs174675 polymorphism of the *COMT* gene and depressive symptoms in adolescents. *COMT* encodes the catechol-O-methyltransferase enzyme, which is involved in the degradation and reuptake of numerous catechols [26]. Its variants have been associated with internalizing problems, such as depression [20] and anxiety [21,22]. 

In the present study, polymorphisms in the *HTR2A* and *FKBP5* genes were not associated with anxiety, depression or chronic pain in Brazilian adolescents. *HTR2A* is responsible for encoding the serotonin receptor, which regulates physiological and cognitive functions [23,52]. This gene has been associated with depression [23] and obsessive-compulsive disorder [53]. The product encoded by *FKBP5* affects glucocorticoid receptor sensitivity and the biological effects of stress and anxiety [19]. Polymorphic variants in *FKBP5* have been associated with a greater risk of depression [24,25], anxiety [25] and post-traumatic stress disorder [24] as well as greater surgical discomfort in third molar extractions [54]. Although we did not find an association between HTR2A and FKBP5 with anxiety, depression and chronic pain in this study, it is important to remember that we evaluated only two polymorphisms within the genes, so it is possible that there are associations in other polymorphisms in these genes with mental disorders. In the present study, we did not find an association between anxiety, depression, chronic pain, and genetic variants with headaches.

It is well known in the literature that psychological aspects such as anxiety and depression are associated with children’s and adolescents’ headaches. Some longitudinal studies with adults have suggested that depression and headaches may present bilateral causes and consequences with the possibility of the influence of shared genetic factors. However, further investigations are needed [55,56,57,58]. In the present study, we did not find an association between anxiety, depression, chronic pain, and genetic variants with headaches.

This study provides useful information on associations between anxiety/depression/chronic pain and OHRQoL/feelings of happiness/polymorphisms in the COMT, HTR2A, and FKBP5 genes in Brazilian adolescents, which are important, considering the limited research on this topic involving adolescents. Although only one polymorphism in COMT was associated with depression, knowledge of genes and factors associated with psychological problems can assist in understanding aspects involved in the etiology of such problems and treatment outcomes, facilitating a personalized approach in clinical practice. As genetic screens are becoming more accessible, identifying whether there is an association between genetic polymorphisms and our outcomes may make it easier to find people who are at increased risk and that can be monitored more closely before it becomes a more serious condition. Since these outcomes are complex and influenced by several issues, we wanted to take a broader view and thereby included patient-centered measures. Studies on mental health issues are important and the authors hope to have shown how psychological conditions can exert a direct impact on the lives of adolescents. 

Our findings should be interpreted in light of some limitations. OHRQoL and feelings of happiness are highly multifactorial elements influenced by several confounding variables, many of which were not included in this study. Additionally, all clinical measures (anxiety, depression and chronic pain) relied on self-report scales. Furthermore, the cross-sectional design does not provide causality. This study also has strengths that should be considered. All instruments were specific and validated for Brazilian adolescents. All data were collected by trained researchers and the kappa coefficient revealed nearly perfect agreement. Further studies with a general sample of adolescents are needed to confirm our findings. Besides, it would be interesting to assess whether serum serotonin levels are associated with polymorphisms, anxiety, depression, and chronic pain. Studies should also evaluate other polymorphisms in the same genes and/or different candidate genes.

## 5. Conclusions

The present findings show that Brazilian adolescents with depression and chronic pain are more likely to considers themselves less happy than others and those with anxiety are more likely to have a negative impact on oral health-related quality of life. Moreover, the rs174675 polymorphism in the *COMT* gene was associated with depressive symptoms in the sample.

## Figures and Tables

**Figure 1 ijerph-20-03321-f001:**
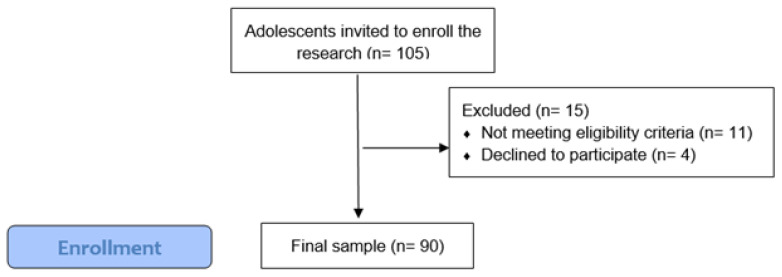
Participants recruitment flowchart.

**Table 1 ijerph-20-03321-t001:** Polymorphisms studied and respective characteristics in general population.

Gene	Polymorphisms	Position	MAF	Altered Base
*COMT*	rs165656	22q11.21	0.465	G > A/C/T
	rs174675	22q11.21	0.339	T > C
*HTR2A*	rs6313	13q14.2	0.403	G > A/C
	rs4941573	13q14.2	0.364	A > C/G
*FKBP5*	rs1360780	6p21.31	0.340	T > A/C
	rs3800373	6p21.31	0.334	C > A/G

Obtained from: http://www.ncbi.nih.gov (accessed on 2 February 2023). MAF: minor allele frequency.

**Table 2 ijerph-20-03321-t002:** Main outcomes frequencies.

Variables	N (%)
Anxiety	
Present	38 (42.7)
Absent	51 (57.3)
Depression	
Present	43 (48.3)
Absent	46 (51.7)
Chronic Pain	
Present	26 (28.9)
Absent	64 (71.1)

Category sums less than N = 90 are due to missing data.

**Table 3 ijerph-20-03321-t003:** Univariate logistic regression models for variables associated with chronic pain, depression and anxiety in adolescents.

Predictors	Without Chronic Pain n (%)	With Chronic Pain n (%)	*p*-Value *	Without Depression n (%)	With Depression n (%)	*p*-Value *	Without Anxiety n (%)	With Anxiety n (%)	*p*-Value *
Sex			0.426			0.001			0.062
Male	33 (75.0)	11 (25.0)		30 (69.8)	13 (30.2)		29 (67.4)	14 (32.6)	
Female	31 (67.4)	15 (32.6)		16 (34.8)	30 (65.2)		22 (47.8)	24 (52.2)	
Regular headache			0.103			0.315			0.225
Yes	16 (88.9)	2 (11.1)		11 (64.7)	6 (35.3)		13 (76.5)	4 (23.5)	
No	7 (63.6)	4 (36.4)		5 (45.5)	6 (54.5)		6 (54.5)	5 (45.5)	
SHS			0.016			0.002			0.04
25th percentile	16.5	15		17	16		17	15	
50th percentile	19	17		20	18		20	18	
75th percentile	22	20		22	20		22	20	
OHRQoL			0.001			0.008			0.001
25th percentile	1	6.75		1	4		2	5.5	
50th percentile	4	11.5		4.5	7		4	11	
75th percentile	7.75	18.25		9.75	15		7	15.75	
*COMT* (rs174675) Dominant									
TT	0 (0.0)	0 (0.0)	#	0 (0.0)	0 (0.0)	^#^	0 (0.0)	0 (0.0)	#
CC + CT	63 (70.8)	26 (29.2)		46 (52.3)	42 (47.7)		50 (56.8)	38 (43.2)	
*COMT* (rs174675) Recessive									
*CC*	46 (66.7)	23 (33.3)	0.112	33 (47.8)	36 (52.2)	0.112	39 (56.5)	30 (43.5)	0.915
*TT + CT*	17 (85.0)	3 (15.0)		13 (68.4)	6 (31.6)		11 (57.9)	8 (42.1)	
*COMT* (rs165656) Dominant									
GG	16 (66.7)	8 (33.3)	0.604	10 (43.5)	13 (56.5)	0.326	13 (56.5)	10 (43.5)	0.973
GC + CC	47 (72.3)	18 (27.7)		36 (55.4)	29 (44.6)		37 (56.9)	28 (43.1)	
*COMT* (rs165656) Recessive									
CC	12 (57.1)	9 (42.9)	0.116	11 (52.4)	10 (47.6)	0.991	13 (61.9)	8 (38.1)	0.59
GG + GC	51 (75.0)	17 (25.0)		35 (52.2)	32 (47.8)		37 (55.2)	30 (44.8)	
*HTR2A* (rs6313) Dominant									
GG	15 (68.2)	7 (31.8)	0.757	12 (54.5)	10 (45.5)	0.805	13 (59.1)	9 (40.9)	0.804
AA + AG	48 (71.6)	19 (28.4)		34 (51.5)	32 (48.5)		37 (56.1)	29 (43.9)	
*HTR2A* (rs6313) Recessive									
*AA*	10 (58.8)	7 (41.2)	0.228	7 (41.2)	10 (58.8)	0.308	10 (58.8)	7 (41.2)	0.853
*AG + GG*	53 (73.6)	19 26.4)		39 (54.9)	32 (45.1)		40 (56.3)	31 (43.7)	
*HTR2A* (rs4941573) Dominant									
AA	20 (74.1)	7 (25.9)	0.653	17 (63.0)	10 (37.0)	0.182	18 (66.7)	9 (33.3)	0.215
AG +GG	43 (69.4)	19 (30.6)		29 (47.5)	32 (52.5)		32 (52.5)	29 (47.5)	
*HTR2A* (rs4941573) Recessive									
*GG*	9 (52.9)	8 (47.1)	0.072	6 (35.3)	11 (64.7)	0.119	9 (52.9)	8 (47.1)	0.719
*AG + AA*	54 (75.0)	18 (25.0)		40 (56.3)	31 (43.7)		41 (57.7)	30 (42.3)	
*FKBP5* (rs1360780) Dominant									
TT	9 (81.8)	2 (18.2)	0.39	8 (72.7)	3 (27.3)	0.147	8 (72.7)	3 (23.7)	0.255
CC + CT	54 (69.2)	24 (30.8)		38 (49.4)	39 (50.6)		42 (54.5)	35 (45.5)	
*FKBP5* (rs1360780) Recessive									
*CC*	26 (65.0)	14 (35.0)	0.278	20 (51.3)	19 (48.7)	0.868	23 (59.0)	16 (41.0)	0.716
*CT + TT*	37 (75.5)	12 (24.5)		26 (53.1)	23 (46.9)		27 (55.1)	22 (44.9)	
*FKBP5* (rs3800373) Dominant									
CC	7 (70.0)	3 (30.0)	0.954	7 (70.0)	3 (30.0)	0.233	8 (80.0)	2 (20.0)	0.116
AA + AC	56 (70.9)	23 (29.1)		39 (50.0)	39 (50.0)		42 (53.8)	36 (46.2)	
*FKBP5* (rs3800373) Recessive									
AA	28 (65.1)	15 (34.9)	0.255	21 (50.0)	21 (50.0)	0.683	22 (52.4)	20 (47.6)	0.422
AC + CC	35 (76.1)	11 (23.9)		25 (54.3)	21 (45.7)		28 (60.9)	18 (39.1)	

* Person’s chi-square test/Values different from 90 are due to missing data. # The *p* value was not computed because there were blank boxes.

**Table 4 ijerph-20-03321-t004:** Multivariate logistic regression models for anxiety in adolescents.

Predictor Variables	*p*-Value	Adjusted OR	95% CI
Subjective happiness scale	0.106	0.895	0.78–1.02
OHIP	0.004	1.093	1.03–1.16
*FKBP5* Polymorphism in dominant A rs3800373			
No (ref)			
Yes	0.127	0.238	0.04–1.51

**Table 5 ijerph-20-03321-t005:** Multivariate logistic regression model for depression in adolescents.

Predictors	*p*-Value	Adjusted OR	95% CI
Subjective Happiness Scale	0.032	0.862	0.75–0.99
OHIP	0.097	2.366	0.85–6.55
*COMT* Polymorphism rs174675 recessive (CC vs. CT + TT)			
No (ref)			
Yes	0.040	0.268	0.08–0.94

**Table 6 ijerph-20-03321-t006:** Multivariate logistic regression model for chronic pain in adolescents.

Predictors	*p*-Value	Adjusted OR	95% CI
Subjective Happiness Scale (total score)	0.042	0.86	0.74–1.00
Sex			
Male (ref)			
Female	0.70	1.24	0.41–3.79
Regular headache			
No (ref)			
Yes	0.57	1.37	0.47–3.97
*COMT* Polymorphism rs165656 dominant (GG vs. GC + CC)			
No (ref)			
Yes	0.188	2.18	0.68–6.96
*COMT* Polymorphism rs174675 recessive (CC vs. CT + TT)			
No (ref)			
Yes	0.249	0.43	0.10–1.80
*HTR2A* Polymorphism rs4941573 dominant (AA vs. AG + GG)			
No (ref)			
Yes	0.072	3.05	0.90–10.32

## Data Availability

Not applicable.

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
