# Peer review of "Associations between Anxiety, Depression, Chronic Pain and Oral Health-Related Quality of Life, Happiness, and Polymorphisms in Adolescents’ Genes"

_ijerph, 2023, doi:10.3390/ijerph20043321_

Round 1

Reviewer 1 Report (Previous Reviewer 1)

The authors corrected what I suggested.

This manuscript is a resubmission of an earlier submission. The following is a list of the peer review reports and author responses from that submission.

Round 1

Reviewer 1 Report

Dear Authors,
This is a well-developed article on an interesting topic for today’s dentistry. However, some issues need to be addressed before considering it for publication:

2. Materials and Methods

- More information on the study design and sample could be provided. In the results, you mentioned that 105 adolescents were invited to participate in this study. Which test did you use for sample size calculation? Describe the way you selected participants from dental offices. Did you include all patients who visited the clinic except those with syndromes and cognitive problems? Describe more the inclusion criteria and exclusion criteria and reasons for dropping out. You can use the CONSORT flow diagram.

- More information about the way SHS and others tests were administered, during face-to-face interviews.

- More information about SHS?

3. Results

- Explain “Table 2. Main outcomes frequencies.” Which test is associated with these results?

- In Materials and methods you mentioned “The adolescents also answered a questionnaire addressing demographic and health related characteristics.”  You do not have a link to these questions in the results. What demographic and health characteristics did you examine? Were there any correlations with other factors from the tests you examined?

Reviewer 2 Report

Very good and ambitious article, subject very interesting for the reader. The article is written in the correct language, the sections of the article are described correctly, minor typographical errors - should be corrected.
The methodology of the research project was developed correctly.
Rich literature, mostly from recent years. References 5, 6, 7 and 24 are missing in the text, please complete.
Please specify the criteria for inclusion and exclusion from the study

Reviewer 3 Report

Associations between anxiety, depression, chronic pain and 2 oral health-related quality of life, happiness, and polymorphisms in adolescents’ genes

The manuscript has strong methodology and clear results, there are few suggestions to improve the clinical impact and the readers understanding of the paper that could be added:

1)    The length of the introduction and the literature review is not appropriate, further elaboration is needed by the author particularly:

·      The OHRQoL, SHS, the COMT, 59 HTR2A, and FKBP5 genes.

·      Why was the OHIP 14 was used instead of other forms.

·      What are the clinical implications of this study.

2)    For the result section:

a.     Table 1 needs further explanation.

b.    What do you mean by “Depression was significantly associated with sex”?

c.     Is there a reason why the subjects have an impaired OHRQoL? Do they have pain complaints?

d.    How was the presence of chronic pain assessed?

e.     Tables 3 & 6 contain the domain “regular headache”, however there was no explanation of that anywhere else in the manuscript. Could you please elaborate?

f.      Also, Table 3 shows that subjects with chronic pain have less headaches, it does not make sense unless you have an explanation. Please explain. For less confusion you could include the results for the chronic pain only.

3)    For the discussion section:

a.      Association does not equal correlation or causation; that should be included as a limitation.